

# Effects of Rotation and Topography on Internal Solitary Waves Governed by the Rotating-Gardner Equation

Karl R. Helfrich[1,*] and Lev Ostrovsky[2,*]

[1]Department of Physical Oceanography, Woods Hole Oceanographic Institution, Woods Hole, MA USA
[2]University of Colorado Boulder, Boulder, Colorado, and University of North Carolina at Chapel Hill, Chapel Hill, NC USA
[*]The authors contributed equally to this work.

**Correspondence:** Lev Ostrovsky (lev.ostrovsky@gmail.com)

**Abstract.** Nonlinear oceanic internal solitary waves are considered under the influence of the combined effects of saturating nonlinearity, the earth's rotation, and horizontal depth inhomogeneity. Here the basic model is the extended Korteweg–de Vries equation that includes both quadratic and cubic nonlinearity (the Gardner equation) with additional terms incorporating slowly varying depth and weak rotation. The complicated interplay between these different factors is explored using an approximate

adiabatic approach and then through numerical solutions of the governing variable-depth, rotating Gardner model. These results are also compared to analysis in the Korteweg–de Vries limit to highlight the effect of the cubic nonlinearity. The study explores several particular cases considered in the literature that included some of these factors to illustrate limitations. Solutions are made to illustrate the relevance of this extended Gardner model for realistic oceanic conditions.

## 1  Introduction

Oceanic internal waves are an important class of nonlinear wave processes. In particular, the internal solitary waves (ISW) are the most ubiquitous type of solitons in the natural environment. These waves often propagate for long distances over several inertial periods, and the effect of Earth's background rotation is potentially significant (e.g. Farmer et al., 2009; Grimshaw et al., 2014; Helfrich, 2007; Ostrovsky and Helfrich, 2019). The large ISWs in the South China Sea are prominent examples (Zhao and Alford, 2006; Alford et al., 2010). There are also numerous remote sensing images throughout the coastal oceans that show

multiple wave packets (e.g. Jackson, 2004), indicating that the ISWs persist over periods longer than the local inertial period. It is also known that rotation destroys internal solitons due to resonant radiation of inertia-gravity waves (terminal damping; see Grimshaw et al., 1998a). The theoretical modeling of such processes often uses the rotation-modified Korteweg–de Vries equation (rKdV) derived for nonlinear waves in rotating ocean (Ostrovsky, 1978). In application to the oceanic conditions, this equation may need additional modifications, specifically for the variable depth along the propagation path. This specific was

considered in Grimshaw et al. (2014) and Ostrovsky and Helfrich (2019).

Another important feature of oceanic solitons is that in many cases they are strongly nonlinear so that the Korteweg–de Vries approximation (KdV) involving only the quadratic nonlinearity is inapplicable (e.g. Apel et al., 2007; Helfrich and Grimshaw, 2008; Ostrovsky and Grue, 2003; Ostrovsky and Irisov, 2017). A better approximation of the real processes can be given by the Gardner equation, that extends KdV by adding the cubic nonlinearity. Although it is formally applicable for small nonlinearity





as well, it preserves the qualitative features of strongly nonlinear waves such as the existence of a limiting soliton amplitude at which its length infinitely increases, whereas a KdV soliton amplitude can unlimitedly increase with its width tending to zero. Moreover, in some cases the Gardner equation gives a good quantitative approximation for very strong solitons, beyond the formal limits of its applicability (Stanton and Ostrovsky, 1998). There are numerous publications using the Gardner equation in applications to oceanic waves including its extension for the rotating ocean (Holloway et al., 1999; Obregon et al., 2018;

Talipova et al., 2015).

In this paper we make the next step by adding a sloping bottom effect to the Gardner model with rotation. Correspondingly, the results of Grimshaw et al. (2014) and Ostrovsky and Helfrich (2019) are significantly modified. Since the Gardner solitons behave differently from the KdV solitons upon propagation, the growth of their amplitude is limited, whereas their length can grow unlimitedly if the depth variation allows that. In particular, we discuss an interplay between the effects of nonlinearity,

rotation, and inhomogeneity. Some realistic estimates are also given.

## 2   Rotating-Gardner equation

A standard model for the evolution of large-amplitude oceanic internal solitary waves is the rotating-Gardner (rG), or extended-KdV equation with rotation and variable depth $h(x)$ (Holloway et al., 1999)

$$\frac{\partial}{\partial x}\left[\frac{\partial \eta}{\partial t} + \left(c + \alpha\eta + \nu\eta^2\right)\frac{\partial \eta}{\partial x} + \beta\frac{\partial^3 \eta}{\partial x^3} + \frac{c}{2}\frac{Q_x}{Q}\eta\right] = \gamma\eta. \tag{1}$$

Here the wave amplitude function $\eta(x,t)$ depends on the horizontal position $x$ and time $t$. The linear long wave phase speed $c$ is found from an eigenvalue problem for the structure function $\Phi(z)$ of a specified vertical mode. Both $c$ and $\Phi(z)$ are slow functions of $x$. The $x$-dependent coefficients $\alpha$, $\nu$, $\beta$, $Q$, and $\gamma$ of the quadratic nonlinear, cubic nonlinear, non-hydrostatic, inhomogeneous, and rotation terms, respectively, are found as integrals over the depth of $\Phi$ or $\Phi'$, and the background stratification $\bar{\rho}(z)$ and current $\bar{u}(z)$. They can be found in numerous publications (e.g. Holloway et al., 1999; Grimshaw et al., 2004)

and are summarized briefly in Appendix I. When considering a spatially inhomogeneous situation it is advantageous to switch from the $(x,t)$ system to the $(s,x)$ system, where

$$s = \int\limits_0^x \frac{dx}{c} - t. \tag{2}$$

If additionally $\zeta = Q^{1/2}\eta$ is introduced, then (1) becomes

$$\frac{\partial}{\partial s}\left[\frac{\partial \zeta}{\partial x} + \frac{\alpha}{c^2 Q^{1/2}}\zeta\frac{\partial \zeta}{\partial s} + \frac{\nu}{c^2 Q}\zeta^2\frac{\partial \zeta}{\partial s} + \frac{\beta}{c^4}\frac{\partial^3 \zeta}{\partial s^3}\right] = \gamma\zeta. \tag{3}$$

We note that here $\zeta^2 = Q\eta^2$ is the wave action flux. Equation (3) can be shown to have two conserved quantities

$$M = \int \zeta\, ds \quad \text{and} \quad E = \int \zeta^2\, ds, \tag{4}$$

where the integrals are over the full $s$ domain (infinite or periodic). These are, respectively, mass and energy related quantities as discussed further in the next section.





Additionally, any initial condition to (3) with $\gamma \neq 0$ must satisfy the zero mass requirement (Ostrovsky, 1978)

$$\int\limits_{T/2}^{-T/2} \zeta(s,0)ds = 0, \tag{5}$$

where $s = -t$ at $x = 0$, and $T$ is the length of the time domain.

In the absence of rotation, $\gamma = 0$, (3) reduces to the Gardner equation that has the solitary wave solutions

$$\zeta = \frac{a}{1 + B\cosh[\sigma(s - \kappa x)]}, \tag{6}$$

described by the parameter $B$. Here

$$a = \frac{Q^{1/2}\alpha}{\nu}(B^2 - 1), \quad \sigma^2 = \frac{c^2\alpha a}{6\beta Q^{1/2}}, \quad \kappa = \frac{\beta\sigma^2}{c^4}. \tag{7}$$

The amplitude, $A$, in terms of $\eta$ ($= \zeta Q^{-1/2}$) is

$$A = \frac{aQ^{-1/2}}{1 + B} = \frac{\alpha}{\nu}(B - 1). \tag{8}$$

From the mass constraint (5), a solitary wave initial condition requires the addition of a constant pedestal,

$$d = -\frac{1}{T}\int\limits_{T/2}^{-T/2} \zeta(s,0)\,ds,$$

with $\zeta$ given by (6).

There are three families of steady solitary wave solutions given by (6)-(8) (Grimshaw et al., 1999). When $\nu < 0$, solitary wave solutions require $0 < B < 1$ and have polarity $\alpha A > 0$. They approach the sech$^2$ KdV solitary wave as $B \to 1$ and as $B \to 0$ the solution approaches the maximum amplitude, $A_{max} = -\alpha/\nu$, flat-top wave. When $\nu > 0$, solitary wave solutions require $B^2 > 1$, and there are two solution branches. For $B > 1$, $\alpha A > 0$ and there is no limit on the wave amplitude. The sech$^2$ KdV wave is recovered as $B \to 1$ from above and for $B \gg 1$ the solution approaches the sech solution of the modified KdV equation (i.e. the Gardner equation with $\alpha = 0$). The third branch occurs for $B < -1$ with the wave polarity $\alpha A < 0$. In this case, solitary wave amplitude has a minimum amplitude $A_{min} = -2\alpha/\nu$ obtained as $B \to -1$ from below. This limiting wave has an algebraic structure and for $B \ll -1$ the solution again approaches the sech wave of modified KdV equation. For $\nu > 0$ there are also localized pulsating traveling wave solutions (breathers) that have total negative mass between zero and the mass of the limiting solitary wave at $B = -1$ (Pelinovsky and Grimshaw, 1997).

## 3  Adiabatic evolution of solitary waves

Assuming that $\zeta \to 0$ for $|s| \to \infty$ and integrating (3) gives

$$\frac{\partial\zeta}{\partial x} + \frac{\alpha}{c^2Q^{1/2}}\zeta\frac{\partial\zeta}{\partial s} + \frac{\nu}{c^2Q}\zeta^2\frac{\partial\zeta}{\partial s} + \frac{\beta}{c^4}\frac{\partial^3\zeta}{\partial s^3} = -\gamma\int\limits_s^\infty \zeta ds'. \tag{9}$$





Multiplication by $\zeta$ and integration from $-\infty$ to $\infty$ gives

$$\frac{d}{dx}\left(\int_{-\infty}^{\infty}\zeta^2 ds\right) = -2\gamma\int_{-\infty}^{\infty}\left(\int_{s}^{\infty}\zeta ds'\right)ds = -\gamma\left(\int_{-\infty}^{\infty}\zeta ds\right)^2 \tag{10}$$

Solution of the rG equation (9) implies that the right side of (10) is zero and $E$ is a constant. However, progress can be made if we start with a solitary wave from (6), assume that the inhomogeneous and rotational effects are very weak such that the evolving wave remains a solitary wave, but with slowly varying amplitude, and take the limits of integration to contain only the evolving solitary wave. With the solitary wave solution (6) written as

$$\zeta = a\mathcal{F}(y), \quad \mathcal{F} = \frac{1}{1+B\cosh(y)}, \quad y = \sigma s \tag{11}$$

(9) gives

$$\frac{d}{dx}\left(\frac{a^2}{\sigma}\mathcal{I}_2\right) = -\gamma\frac{a^2}{\sigma^2}\mathcal{I}_1^2 \tag{12}$$

where

$$\mathcal{I}_n = \int_{-\infty}^{\infty}\mathcal{F}^n dy, \quad n = 1, 2. \tag{13}$$

The result, (12), is a statement for the variation of the energy,

$$E_w = \frac{a^2}{\sigma}\mathcal{I}_2, \tag{14}$$

of the slowly evolving solitary wave. In the absence of rotation $E_w$ is conserved; however, the wave mass,

$$M_w = \frac{a}{\sigma}\mathcal{I}_1, \tag{15}$$

is not necessarily conserved. Changes in $M_w$ are compensated by the formation of a trailing shelf. For example, in the typical case with $\nu < 0$ of a solitary wave approaching a point of polarity reversal, $\alpha = 0$, the wave mass increases in magnitude (Grimshaw et al., 1998b). Thus the trailing shelf must have the sign opposite to the wave polarity. When $\nu > 0$, the situation is more subtle, but again any variations in $M_w$ are compensated by a shelf (Grimshaw et al., 1999; Nakoulima et al., 2004). In a homogeneous, rotating environment both $E_w$ and $M_w$ decrease with $x$ and are compensated by the trailing wave radiation (Grimshaw et al., 1998a). With both inhomogeneity and rotation the energy will decrease with $x$, but the wave mass may increase or decrease depending on the interplay between these two effects.

Note that for the KdV equation and with $Q$ defined as in (A.3d), $\rho_0 E_w$ is the solitary wave energy, $\iint pu\,dz\,dt$. Here $p$ and $u$ are the first-order, wave-induced pressure and horizontal velocity fields. With the addition of the cubic nonlinear term in the Gardner equation, $E_w$, is not exactly the wave energy, but is still a good measure of it. The actual wave mass is $Q^{-1/2}M_w$.




### 3.1 Rotating-KdV equation

For later reference we first consider the adiabatic theory for the rotating-KdV equation ($\nu = 0$) from Grimshaw et al. (2014). The solitary wave solution is found from (6) and (8) with $B = 1$ as

$$\zeta = a\mathcal{F}, \qquad \mathcal{F} = \mathrm{sech}^2(y), \qquad \sigma^2 = \frac{c^2 \alpha a}{12\beta Q^{1/2}}, \qquad a = AQ^{1/2}. \tag{16}$$

From (13), $\mathcal{I}_1 = 2$ and $\mathcal{I}_2 = 4/3$, and (12) gives

$$\frac{d}{dx}\left(\frac{4}{3}\frac{a^2}{\sigma}\right) = -4\gamma \frac{a^2}{\sigma^2}. \tag{17}$$

This can be written as

$$\frac{d}{dx}\mathcal{A}^{3/2} = -3\gamma w^2 \mathcal{A}, \qquad \mathcal{A} = wa, \qquad w = \left(\frac{12\beta Q^{1/2}}{\alpha c^2}\right)^{1/3}.$$

When integrated this gives

$$\frac{A}{A_0} = \left(\frac{Q_0}{Q}\right)^{1/2}\frac{a}{a_0} = \left(\frac{Q_0}{Q}\right)^{1/2}\frac{w_0}{w}\left[1 - (w_0 a_0)^{-1/2}\int_0^x \gamma w^2 dx'\right]^2. \tag{18}$$

The 0 subscript indicates the initial value at $x = 0$.

In the absence of rotation, $\gamma = 0$, conservation of wave action gives

$$\frac{A}{A_0} = \left(\frac{Q_0}{Q}\right)^{1/2}\frac{w_0}{w} = \left(\frac{Q_0^2 \beta_0}{\alpha_0 c_0^2}\frac{\alpha c^2}{Q^2 \beta}\right)^{1/3}. \tag{19}$$

For a homogeneous, rotating environment (18) gives

$$\frac{A}{A_0} = \left[1 - \frac{x}{X_{eO}}\right]^2, \qquad X_{eO} = \frac{c}{\gamma}\left(\frac{\alpha A_0}{12\beta}\right)^{1/2}. \tag{20}$$

The KdV solitary wave is completely extinguished by radiation of inertia-gravity waves in the finite distance $X_{eO}$.

### 3.2 Rotating-Gardner equation

When $\nu \neq 0$, the Gardner solitary wave solution (11) and (13) give

$$\mathcal{I}_1 \;\; = \;\; \frac{4}{(B^2 - 1)^{1/2}}\,\mathcal{T}(B), \tag{21}$$

$$\mathcal{I}_2 \;\; = \;\; \frac{2}{B^2 - 1} - \frac{4}{(B^2 - 1)^{3/2}}\,\mathcal{T}(B), \tag{22}$$

where

$$\mathcal{T}(B) = \tan^{-1}\left(\frac{B - 1}{\sqrt{B^2 - 1}}\right) = \mathrm{sgn}(B)\,\tan^{-1}\sqrt{\frac{B - 1}{B + 1}}. \tag{23}$$





Substituting (7), (21) and (22) into (12) gives

$$\frac{d}{dx}\left[\left(\frac{24\beta\alpha^2 Q^2}{c^2\nu^3}\right)^{1/2}\left[(B^2-1)^{1/2}-2\mathcal{T}(B)\right]\right]=-96\frac{\gamma\beta Q}{c^2\nu}\mathcal{T}^2(B). \tag{24}$$

This equation can be integrated to obtain $B(x)$, hence $A(x)$ from (8). Since $B \to 1$ as $\nu \to 0$, the equation remains regular for cases with $\nu(x)$ changing sign. Note also that it remains real for $\nu < 0$ and $0 < B < 1$ since $\tan^{-1}(iy) = i\tanh^{-1}(y)$.

While it is not necessary to make the sign of $\nu$ explicit in integrating (24), for $\nu < 0$ ($0 < B < 1$), it can be written as

$$\frac{\sqrt{1-B^2}}{B}\frac{dB}{dx} = \frac{1}{2}\frac{d}{dx}\left[\ln\left(\frac{\beta Q^2\alpha^2}{c^2|\nu|^3}\right)\right]\left(2\tanh^{-1}\sqrt{\frac{1-B}{1+B}}-\sqrt{1-B^2}\right)$$
$$+8\gamma\left(\frac{6\beta|\nu|}{c^2\alpha^2}\right)^{1/2}\left(\tanh^{-1}\sqrt{\frac{1-B}{1+B}}\right)^2. \tag{25}$$

and for $\nu > 0$ ($B^2 > 1$)

$$\frac{\sqrt{B^2-1}}{B}\frac{dB}{dx} = -\frac{1}{2}\frac{d}{dx}\left[\ln\left(\frac{\beta Q^2\alpha^2}{c^2\nu^3}\right)\right]\left(\sqrt{B^2-1}-2\tan^{-1}\left(\frac{B-1}{\sqrt{B^2-1}}\right)\right)$$
$$-8\gamma\left(\frac{6\beta\nu}{c^2\alpha^2}\right)^{1/2}\left[\tan^{-1}\left(\frac{B-1}{\sqrt{B^2-1}}\right)\right]^2. \tag{26}$$

As discussed above, in a non-rotating system the right-hand side of (24) is zero, and the conservation of wave energy gives

$$E_{w0} = \left(\frac{24Q^2\alpha^2\beta}{c^2\nu^3}\right)^{1/2}\left[(B^2-1)^{1/2}-2\mathcal{T}(B)\right], \tag{27}$$

where $E_{w0}$ is a constant evaluated at $x = 0$. This can be solved to give $B(x)$.

Radiation decay in a homogeneous environments (where $c$, $\alpha$, ... are constants) was recently considered by Obregon et al.
(2018). However, there is an error in their development that renders their quantitative results incorrect. Thus the homogeneous radiation decay problem is briefly redeveloped here. The origin of their error is discussed in Appendix B.

In a homogeneous environment the first term on the right-hand-side of (25) or (26) is zero. Then the distance, $X_{eG}$, to complete radiation decay in the rG equation is found by integration of (25) or (26), with the first term on the right side set to zero, from $x = 0$ to $X_{eG}$, where $B = B_0$ and $B_e$, respectively. When $A_0\alpha > 0$, $B_0 > 0$, the solitary wave decays from an initial
amplitude $A_0 = \alpha\nu^{-1}(B_0 - 1)$ to zero when $B_e = 1$, regardless of the sign of $\nu$. While for $\nu > 0$ and $A_0\alpha < 0$, $B_0 < -1$ and the solitary wave decay can be followed only to the limiting amplitude, $A_{lim} = -2\alpha/\nu$, where $B_e = -1$. These considerations give

$$\frac{X_{eG}}{X_{eO}} = \frac{\sqrt{2}}{8}\begin{cases} \mathcal{I}_B(B_0,1)|B_0-1|^{-1/2}, & B_0 > 0, \\ \\ \mathcal{I}_B(B_0,-1)(1-B_0)^{-1/2}, & B_0 < -1, \end{cases} \tag{28}$$

where

$$\mathcal{I}_B(B_0,B_e) = \begin{cases} \int_{B_0}^{B_e}\frac{\sqrt{1-B^2}}{B}\left(\tanh^{-1}\sqrt{\frac{1-B}{1+B}}\right)^{-2}dB, & \nu < 0, \\ \\ -\int_{B_0}^{B_e}\frac{\sqrt{B^2-1}}{B}\left(\tan^{-1}\sqrt{\frac{B-1}{B+1}}\right)^{-2}dB, & \nu > 0. \end{cases} \tag{29}$$

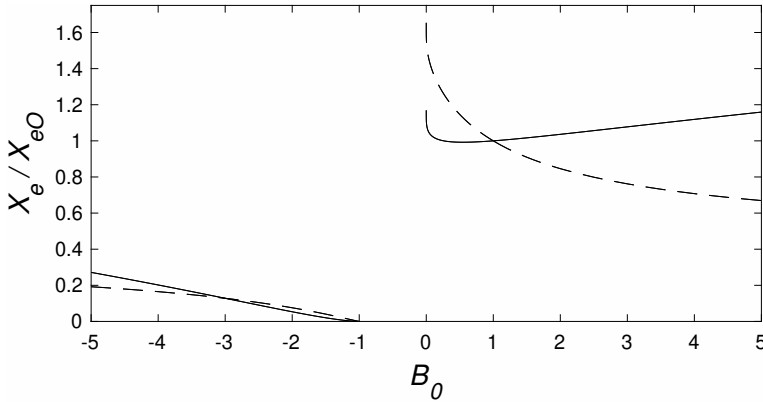

**Figure 1.** $X_{eG}/X_{eO}$ as a function of $B_0$ for the three Gardner solitary wave regimes: $\nu < 0$ for $0 < B_0 < 1$ and $\nu > 0$ for $B^2 > 1$. The solid lines are computed from (28). The dashed lines show the Obregon et al. (2018) results.

Again, $X_{eO}$ is the rKdV equation decay length (20) evaluated using $|A_0 \alpha|$ when $B_0 < -1$.

Figure 1 shows $X_{eG}/X_{eO}$ versus $B_0$ for all three wave regimes. For $\nu < 0$ where $0 < B_0 < 1$, $X_{eG}/X_{eO} \approx 1$ for $0.2 < B_0 < 1$. There is a slight minimum of 0.9924 at $B_0 = 0.55$. As $B_0 \to 0$ the ratio increases to $X_{eG}/X_{eO} = 1.1842$ at $B_0 = 10^{-14}$ and appears to approach a finite limit as $B_0 \to 0$. For $B_0 > 1$ ($\nu > 0$), $X_{eG}/X_{eO}$ increases monotonically from one with $B_0$, but remains $O(1)$ even for $B_0 = 10$. Similar behavior is found for $B_0 < -1$ ($A_0 \alpha > 0$), although it should be remembered that in this regime the adiabatic theory gives an amplitude only until the limiting wave at $B_e = -1$ is reached. For comparison, the Obregon et al. (2018) results are also shown.

Figure 2 shows examples of normalized wave amplitude $A/A_0$ as functions of $x/X_{eO}$ for several values of $B_0$ in each wave regime. For $\nu < 0$ and $B_0 \gtrsim 0.1$ (Figure 2a) the amplitude decay closely follows the solution (20) for the rKdV equation, but as $B_0$ decreases the initial amplitude decay rate slows. As mentioned above, for $\nu > 0$ and $B_0 < -1$ (Figure 2c) the decay can only be followed until the limiting wave amplitude is reached.

## 4 Comparisons with rG numerical solutions

In this section the adiabatic theory (24) is compared with full numerical solutions of the rG equation (3). Example cases employ spatially uniform stratifications and inhomogeneous effects are introduced by variations in the total water depth. The numerical solutions of (3) are found using a de-aliased pseudo-spectral scheme in $s$ with a third-order Runge-Kutta integration in $x$. The relations for the coefficients, $\alpha(x)$, $\nu(x)$, ..., are given in Appendix A.



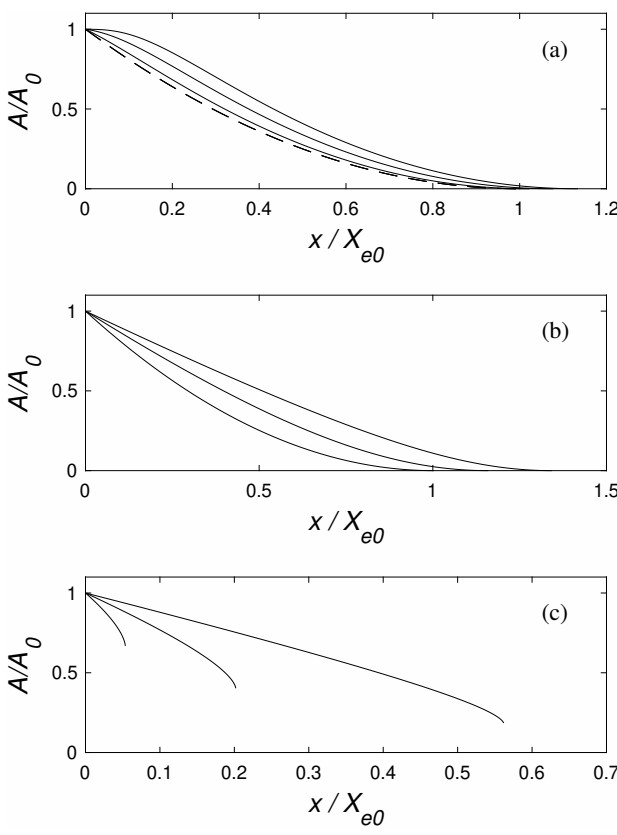

**Figure 2.** $A/A_0$ versus $x/X_{eO}$ for homogeneous conditions. a) $\nu < 0$ for $B_0 = [10^{-1}, 10^{-2}, 10^{-4}]$ (solid lines from left to right). The dashed line is the rKdV equation solution (20). b) $\nu > 0$ and $B_0 = [1.1, 5, 10]$ (from left to right). c) $\nu > 0$ and $B_0 = [-2, -5, -10]$ (from left to right).

### 4.1 Rotating, homogeneous evolution

In the homogeneous case where the coefficients $c$, $\alpha$, ... are constants it is convenient to reduce (3) to an equation with only one parameter by introducing the scaling

$\quad u = \dfrac{\zeta}{U}, \quad \tau = \dfrac{s}{T}, \quad \xi = \dfrac{x}{L}$

where

$$U = \frac{\alpha}{|\nu|}, \quad T = \left(\frac{\beta|\nu|}{c^2\alpha^2}\right)^{1/2}, \quad L = \frac{c}{\alpha^3}\left(\beta|\nu|^3\right)^{1/2}.$$

This change of variables gives

$$\frac{\partial}{\partial\tau}\left[\frac{\partial u}{\partial\xi} + u\frac{\partial u}{\partial\tau} + \text{sgn}(\nu)u^2\frac{\partial u}{\partial\tau} + \frac{\partial^3 u}{\partial\tau^3}\right] = \epsilon u \quad \text{with} \quad \epsilon = \gamma\beta\frac{\nu^2}{\alpha^4}. \tag{30}$$


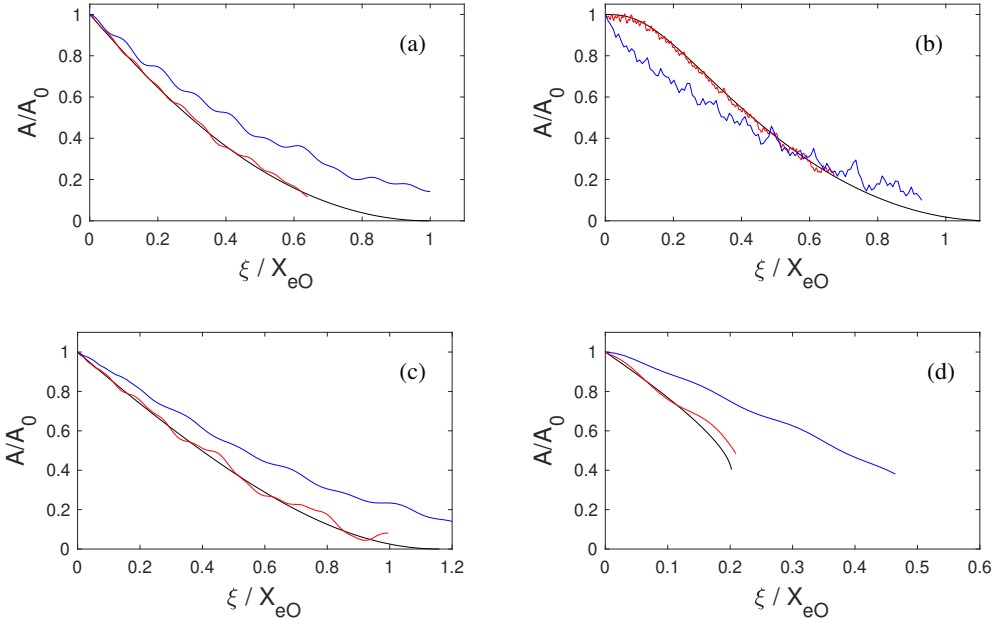

**Figure 3.** Comparison of the rG adiabatic radiation decay theory for homogeneous conditions (black lines) and rG numerical solutions for a) $B_0 = 0.3$, b) $B_0 = 10^{-4}$, c) $B_0 = 5$, and d) $B_0 = -4$. In (a) and (b) red (blue) indicates numerical solutions of the rG equation (30) for $\epsilon = 2.5 \times 10^{-5}$ ($2.5 \times 10^{-3}$). In (c) and (d) red (blue) indicates $\epsilon = 2.5 \times 10^{-3}$ (0.25).

The solitary wave solution (6)-(8) and the adiabatic radiation decay results carry through after taking $c = Q = \alpha = \beta = 1$, $\nu = \pm 1$, and $\gamma = \epsilon$. In these variables $X_{eO} = \epsilon^{-1}(|A_0|/12)^{1/2}$ and $A_0 = \text{sgn}(\nu)(B_0 - 1)$.

Figure 3a-d shows comparison of the scaled amplitude $A/A_0$ versus $\xi/X_{eO}$ from the adiabatic theory and full numerical solutions of the rG equation (30) with $B_0 = [0.3, 10^{-4}, 5, -4]$, respectively and $2.5 \times 10^{-5} \le \epsilon \le 0.25$ as indicated. (The small oscillations in the amplitude are due to the periodic boundary conditions used in the numerical solutions that allowed the radiated waves to re-enter the domain upstream of the solitary wave.) For $B_0 = 10^{-4}$ and 0.3 ($\nu < 0$) the agreement between the adiabatic theory and the rG solutions is quite good for $\epsilon = 2.5 \times 10^{-5}$. However, for $\epsilon = 2.5 \times 10^{-3}$ there is disagreement. Similarly, for $B_0 = -4$ and 5 the agreement also declines with increasing $\epsilon$, although in these examples the agreement for $\epsilon = 2.5 \times 10^{-3}$ is good. Increasing $\epsilon$ generally results in slower amplitude decay. The exception is $B_0 = 10^{-4}$, where the initial decay is more rapid. This rapid decay for near-maximal waves was also found by Obregon et al. (2018) which they attributed to a structural instability of large-amplitude, flat-top solitary waves.

Figure 3c with $B_0 = 5$ clearly indicates that the decay distance is in much closer agreement with the prediction of $X_e/X_{eO} = 1.16$ from (29) than $X_e/X_{eO} = 0.67$ from Obregon et al. (2018) (see Figure 1).

The complicated evolution of the decaying solitary wave and the trailing radiation is illustrated in Figures 4a and b. Figure 4a is the $B_0 = 10^{-4}$ and $\epsilon = 2.5 \times 10^{-5}$ example in Figure 3b. As the initial solitary wave decays, the trailing radiation itself steepens to form a group of solitary-like waves that also decay by radiation damping. Over larger distances this radiation will


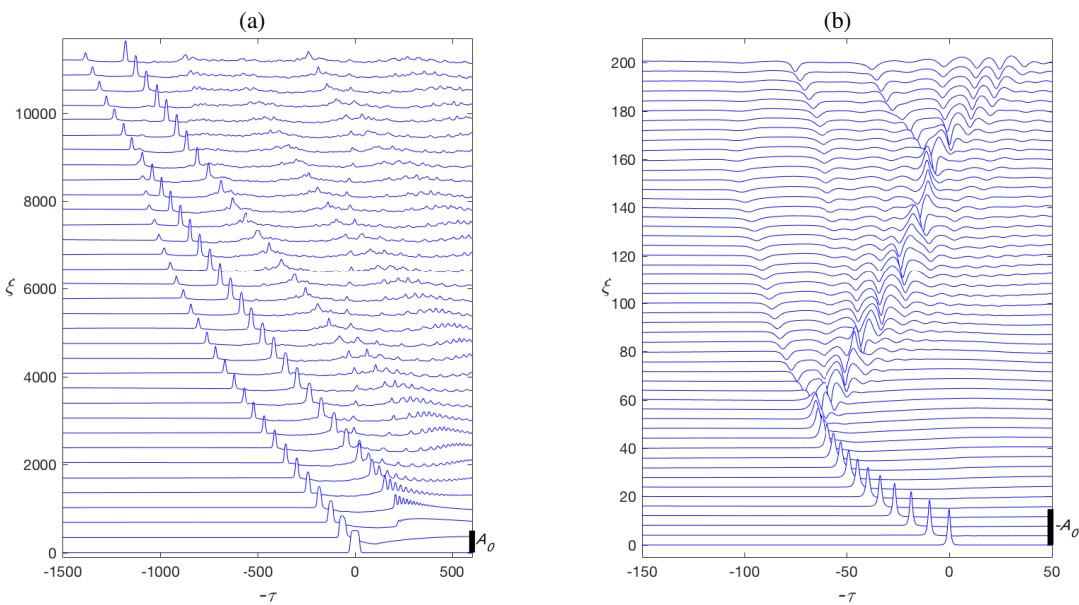

**Figure 4.** Numerical solution of (30) with a) $\epsilon = 2.5 \times 10^{-5}$ and $B_0 = 10^{-4}$ and b) $\epsilon = 2.5 \times 10^{-3}$ and $B_0 = -4$. The solid bar indicates the initial wave amplitude $A_0 = \text{sgn}(\nu)(B_0 - 1)$. Only a portion of the full $\xi$ domain is shown. The locations of each time series are indicated by the vertical axis.

likely organize into one or more nonlinear wave packets as found by Grimshaw and Helfrich (2008) for the rotating-KdV equation and Whitfield and Johnson (2015) in the rotating-Gardner equation.

The example in Figure 4b has $B_0 = -4$ and $\epsilon = 2.5 \times 10^{-3}$ (see Figure 3d). For these parameters the evolution is further complicated since the radiation decay ceases at $\xi \approx 50$ $(= 0.21 X_{eO})$ when the wave amplitude decays to the limiting ampli-

195   tude $A_{lim} = -2\alpha/\nu = -2$ in these scaled variables. The wave then rapidly forms what appears to be a small solitary wave of reversed polarity $(B_0 > 1)$ and a trailing wave packet that has characteristics similar to the breather solutions of the Gardner equation. However, this packet subsequently disintegrates due to rotational effects. The complicated nature of the wave evolution with rotation in homogeneous conditions suggests even more interesting features when both rotation and inhomogeneous effects are active.

200   **4.2   Combined inhomogeneous and rotation effects**

Ostrovsky and Helfrich (2019) showed that for the rKdV equation the competition between extinction by radiation decay and at the point of polarity reversal, $\alpha = 0$, could be characterized by the ratio of the inhomogeneous and rotation terms of (1)

$$C \approx \frac{c}{\gamma} \frac{Q_x}{Q} \frac{\eta_x}{\eta} \sim \frac{c}{\gamma L \Delta} = \frac{X_{eO}}{L}, \tag{31}$$



where $L$ is the length scale over which inhomogeneous term $Q$ varies, say the distance to the $\alpha = 0$ location. $\Delta$ is the solitary wave scale, taken above to be the KdV wave scale $\Delta_{KdV} = (12\beta/A_0\alpha)^{1/2}$. For $C \gg 1$ inhomogeneous effects dominate and conversely rotational decay dominates for $C \ll 1$. Alternatively $\Delta$ might taken to be the Gardner solitary wave scale $\Delta_G = \Delta_{KdV}[2/(1 + B_0)]^{1/2}$. However, for $B_0 > 0$ and not too large, the term in parentheses is $O(1)$ and (31) is a reasonable scaling estimate. One could simply define $C = X_{eG}/L$, but since $X_{eG}/X_{eO} \approx 1$ for $0 < B_0 < 10$ (see Figure 1) this also gives $C$ from (31). The exception is for situations with $B_0 < -1$, since $X_{eG}/X_{eO}$ can be much less than one.

To illustrate the combined effects of inhomogeneity and rotation a two-layer Boussinesq stratification with upper layer depth $h_1$, variable lower layer depth $h_2(x)$, reduced gravity $g' = g(\rho_2 - \rho_1)/\rho_1$, and Coriolis frequency $f$ will be considered first. Here $\rho_1$ and $\rho_2$ are the densities of the upper and lower layers, respectively. Relations for the coefficients $\alpha(x)$, $\nu(x)$, etc. are given in (A.6). Note that $\nu < 0$ for two-layer stratifications. Thus only the $0 < B_0 < 1$ branch of solitary wave solutions is possible. The wave polarity is given by the sign of $\alpha$ with $\alpha < 0$ ($> 0$) for $h_1/h_2 < 1$ ($> 1$).

The bottom slope will be taken constant and the lower layer depth given by

$$\frac{h_2(x)}{h_{20}} = 1 - \left(1 - \frac{h_1}{h_{20}}\right)\frac{x}{L}, \tag{32}$$

where $h_{20} = h_2(0)$ and $L$ is the distance from the origin to the critical point where $\alpha = 0$ (i.e., $h_1 = h_2$). The case $h_1/h_{20} < 1$ corresponds to an initial solitary wave with $A_0 < 0$ propagating from deep to shallower water. Propagation of a positive wave, $A_0 > 0$, from shallow to deeper water occurs for $h_1/h_{20} > 1$. While the adiabatic solutions can be obtained only up to the critical point, $x \leq L$, numerical solutions of the rG equation are found beyond the critical depth. In the deep-to-shallow situation the bottom slope is continued until $h_2 = h_1/2$, beyond which $h_2$ is constant over a shelf region.

The evolution of the wave amplitude $A(x)$ is shown in Figure 5a for a deep to shallow case for a representative oceanographic situation with $h_1 = 50$ m, $h_{20} = 450$ m, $g' = 0.005$ m s$^{-2}$, $f = 10^{-4}$ s$^{-1}$, and $A_0 = -25$ m. For this initial wave $B_0 = 0.764$ so that effects of the cubic nonlinearity are present, but do not dominate initially. Slope lengths $L = 50, 100, 200$, and $400$ km are considered and give $C = 3.46, 1.73, 0.86$, and $0.43$, respectively, from (31). The solid lines show the rotating adiabatic theory and the dashed line shows the $f = 0$ solution (equivalent for all $L$ when plotted against $x/L$). As anticipated from the values of $C$, rotational decay effects increase significantly as $L$ increases with complete, or nearly complete, extinction for the two longer slopes. However, even for $L = 50$ km, rotation causes a noticeable reduction in wave amplitude compared to the non-rotating solution.

Figure 5b shows the wave energy, $E_w(x)$ from (14), for the same parameters. The ratio $E_w/E_{w0}$ is a measure of the fraction of initial wave energy that remains in the evolving solitary wave, with the difference lost to the trailing inertia-gravity wave radiation. Even the shortest slope, $L = 50$ km, where the effects of rotation were relatively weak, more than half of the initial wave energy is lost to inertia-gravity wave radiation by $x/L \approx 0.8$.

The variation of wave mass, $M_w$ from (15), is plotted in Figure 5c. In all cases rotation causes an initial reduction in wave mass, which is compensated in the trailing radiation and emerging shelf (with the same mass sign as the initial wave). The mass goes to zero for the two longer slopes prior to the critical point, while for the two shorter slopes the wave mass increases rapidly as the critical point is approached, similar to the non-rotating result. However, increased slope length (i.e. rotational

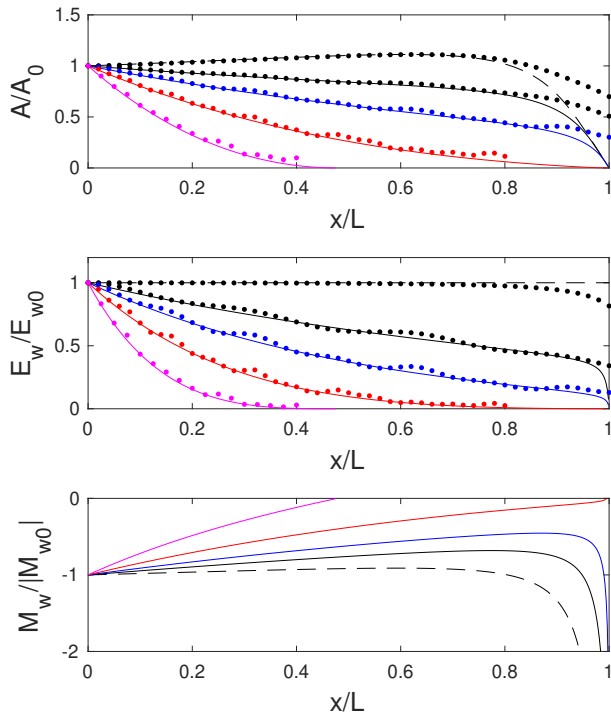

**Figure 5.** Adiabatic theory for wave propagation from deep to shallow in a two-layer system with $h_1 = 50$ m, $h_{20} = 450$ m, $g' = 0.005$ m s$^{-2}$, $f = 10^{-4}$ s$^{-1}$, and $A_0 = -25$ m. $A/A_0$, $E_w/E_{w0}$, and $M_w/|M_{w0}|$ are shown as functions of $x/L$. The solid lines are for $L = 50$ (black), 100 (blue), 200 (red), and 400 km (magenta). The dashed lines are for $f = 0$. The dots are from solutions of (3).

effects) delays this growth with consequences for the magnitude of the trailing shelf and subsequent wave structure transmitted onto the topographic shelf (see below).

Also shown in Figures 5a and b are results from the numerical solutions to the rG equation (3). The energy of the solitary wave in the rG model is found by integrating $Q\eta^2$ in a small region encompassing the solitary wave that does not incorporate appreciable trailing radiation. The agreement is quite good for all the cases, except for $x/L \gtrsim 0.8$ and $L = 50$ and 100, where the amplitude from the rG numerical calculation does not decay as rapidly as the adiabatic model. This is consistent with previous studies without rotation (Grimshaw et al., 1998b, 1999). As the point of polarity reversal is approached the wave

elongates to form a rarefaction, and the trailing shelf with opposite polarity emerges. Figure 5 shows that the disagreement is associated with rapid changes in the wave amplitude and mass. In this region that assumption of slow variation of the inhomogeneity fails.

Two examples of the full rG solutions for $L = 50$ km and 200 km are shown in Figure 6 and the rG solutions at $x/L = 1.2$ for all five cases in Figure 5 are compared in Figure 7. Note that $h_2 = h_1/2$ for $x/L \geq 1.0625$ on the shelf. Again, even for

$L = 50$ km rotation leads to a clear effect on the wave signal transmitted on to the shelf as the leading crest of the (weak) trailing inertia-gravity wave also steepens form a second transmitted wave packet. The second packet is consistent with the

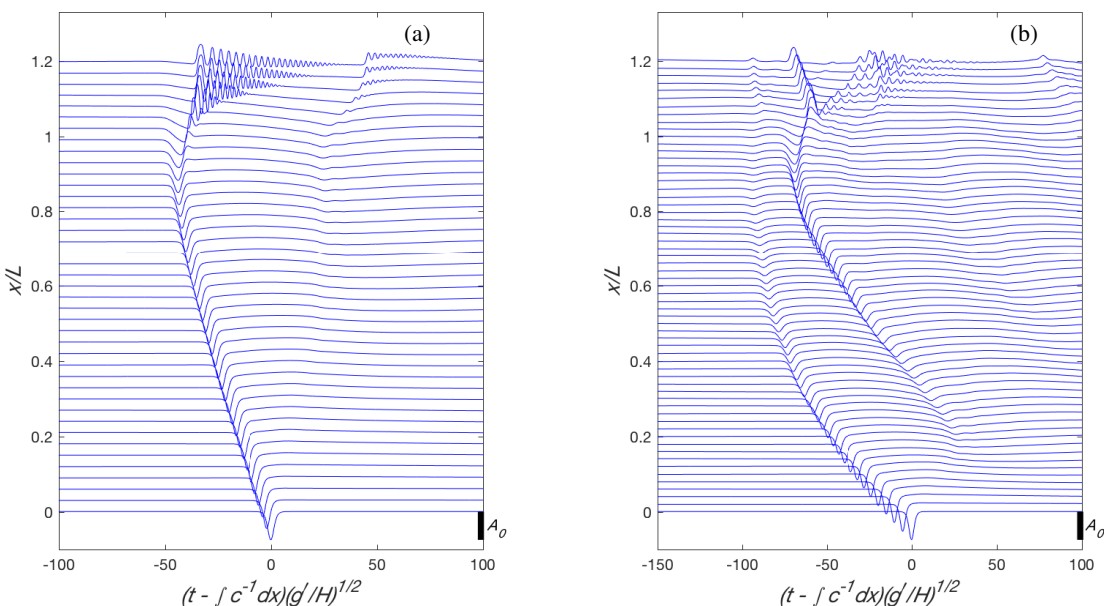

**Figure 6.** rG numerical solutions for $h_1 = 50$ m, $h_{20} = 450$ m, $g' = 0.005$ m s$^{-2}$, $f = 10^{-4}$ s$^{-1}$ and $A_0 = -25$ m. a) $L = 50$ km and b) $L = 200$ km. The wave amplitude $\eta(x,t)$ is shown at $x/L$ as function of normalized shifted time, $-s(g'/H)^{1/2}$, where $H = h_1 + h_{20}$. The initial wave amplitude is indicated by the scale on the lower right of each panel.

breaking criteria obtained by Grimshaw et al. (2012) and similarly noted in Grimshaw et al. (2014). Further increases in rotation effects lead to an additional transmitted packet for $L = 100$ km. For the two longest slopes, $L = 200$ and $400$ km the transmitted signal becomes increasingly disorganized. Figure 6b illustrates the evolution leading to this outcome. In this example the initial

solitary wave is extinguished before the critical point is reached, However, the trailing inertia-gravity wave steepens to produce a solitary wave that is itself scattered through the critical point. However, the interaction of the scattered signal with the trailing radiation gives rise to the disorganization.

In the examples above the cubic nonlinearity was not an essential feature of the evolution. Indeed the wave evolution is qualitatively similar to the rotating-KdV solutions in Figure 2 of Ostrovsky and Helfrich (2019). In Figure 8 another example

with $h_1 = 100$ m, $h_{20} = 200$ m, $g' = 0.01$ m s$^{-2}$, $f = 10^{-4}$ s$^{-1}$, and $L = 40$ km is shown. Initial wave amplitudes $A_0 = -10$ m and $-45$m, corresponding to $B_0 = 0.788$ and $0.0478$, respectively. The larger wave is very close to the limiting amplitude $A_{lim} = -47.06$ m. The competition parameter $C = 4.56$ and $9.68$, respectively. The left column of Figure 8 shows the rG adiabatic solutions and for comparison the right column shows the adiabatic solutions obtained from the rKdV theory (18). Now the differences between the rG and rKdV solutions are substantial, especially for the large wave. The rKdV solution has

this larger wave decaying more slowly than the smaller wave, while the rG model shows just the opposite. Similar to above, rotation, even for this relatively short slope (and hence weak rotational effect), causes significant energy loss in the primary

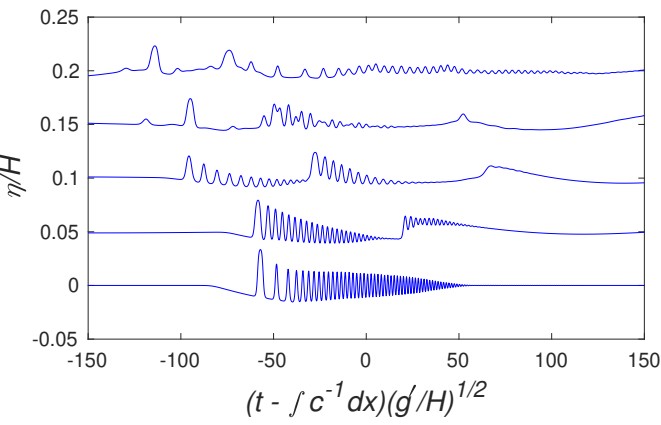

**Figure 7.** $\eta/H$ at $x/L = 1.2$ for the parameters of Figure 5. The bottom curve is the non-rotating case with $L = 50$ km. The top four include rotation and have $L = 50, 100, 200,$ and $400$ km from bottom to top. The curves are offset by 0.05.

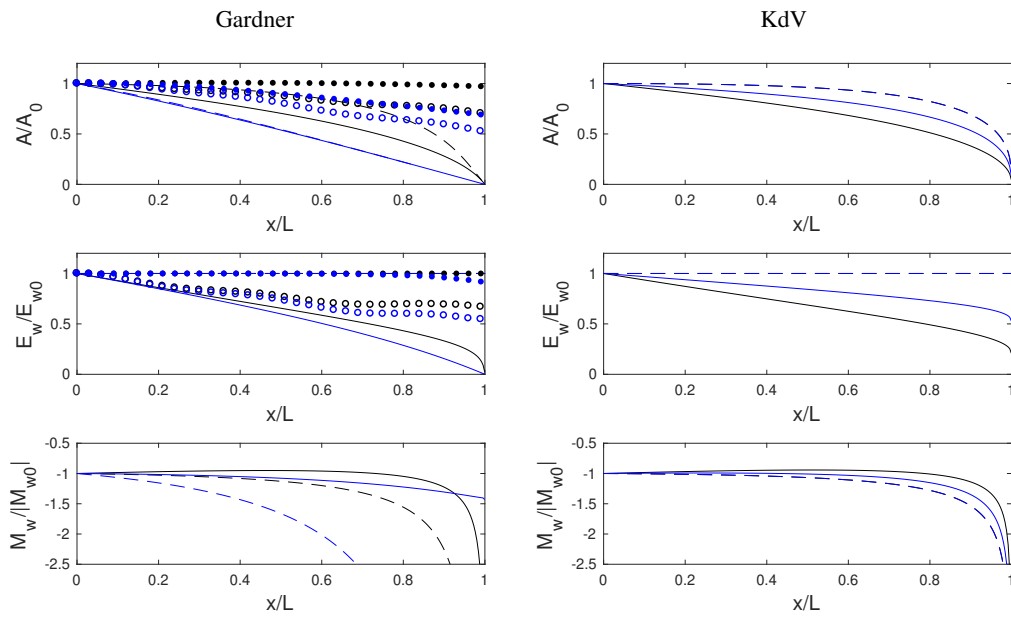

**Figure 8.** Adiabatic theory for wave propagation from deep to shallow in a two-layer system with $h_1 = 100$ m, $h_{20} = 200$ m, $g' = 0.01$ m s$^{-2}$, $f = 10^{-4}$ s$^{-1}$, and $L = 40$ km. The left column shows $A/A_0$, $E_w/E_{w0}$ and $M_w/|M_{w0}|$ for the rotating-Gardner theory. The right column shows the equivalent rotating-KdV solutions. The solid (dashed) lines are with (without) rotation for $A_0 = -10$ m (black) and $-45$ m (blue). The open (solid) symbols are from corresponding numerical solutions of (3) with (without) rotation.

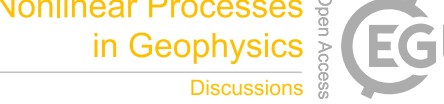

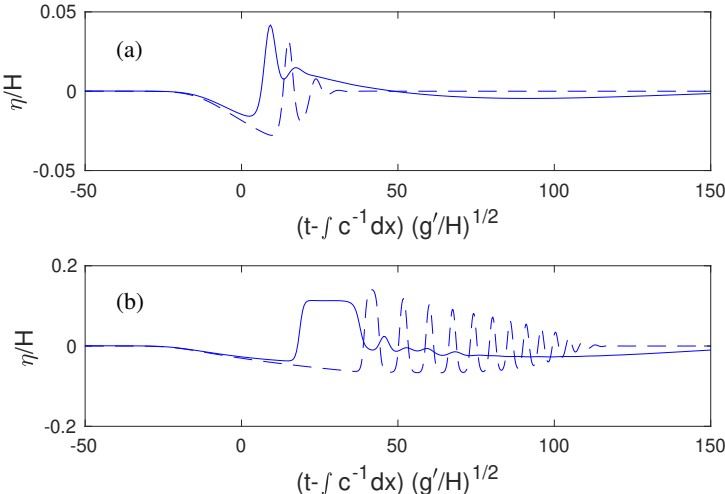

**Figure 9.** $\eta/H$ at $x/L = 1.5$ for the parameters of Figure 8. a) $A_0 = -10$ m and b) $A_0 = -45$ m. The solid (dashed) lines are with (without) rotation.

wave as it climbs the slope. For $A_0 = -45$m, the mass $M_w$ remains finite at the critical point, indicating the generation of a relatively weak trailing shelf. The agreement between the rG adiabatic model and the full rG solutions for $A(x)$ is not very good, although the qualitative prediction that the large wave should decay more rapidly is found in both cases. The origin of

the disagreement is likely due to the lack of separation between the wave scales and scale of the inhomogeneity.

The transmitted signals at $x/L = 1.5$ from the full rG numerical solutions with and without rotation are shown in Figure 9. For both initial wave amplitudes rotation leads to significant changes in the transmitted signal. This is especially true for $A_0 = -45$ m, where the transmitted packet without rotation is replaced by a single, broad wave emerging onto the constant depth shelf with a much weaker trailing signal. However, on the shelf $A_{lim}/H = 0.0745$, so that this leading wave must further

adjust and is also subject to continued rotational decay.

## 5  Concluding Remarks

In this paper we have continued a series of studies of nonlinear internal waves of a moderate amplitude in the shallow, stratified areas of the ocean (see the references in the Introduction). Based on the classical Korteweg-de Vries equation, we added the main factors making the analysis closer to the physical reality: cubic nonlinearity, Earth's rotation, and sloping bottom.

Interplay of these factors makes the problem rather complicated, both physically and mathematically. To better explain the qualitative effect of each of them, first we briefly reproduce the effect of rotation in the medium with quadratic nonlinearity (KdV with rotation), then that with both quadratic and cubic nonlinearities (Gardner with rotation) and, as the main content of this paper, the joint effect of rotation and inhomogeneity in the Gardner equation. The specific qualitative effect of the latter is the limiting soliton amplitude and the corresponding increase of the wavelength so that the topography effect becomes espe-



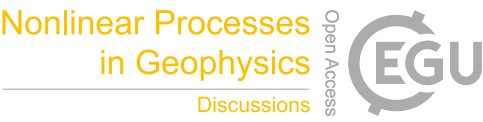

cially important. Along with the approximate adiabatic approach, a direct numerical study of the rG equation was performed and confirmed the adiabatic theory and highlighted its limitations. This combined approach allowed us to demonstrate a rather complicated behavior of shoaling internal solitons. For example, whereas the soliton energy always decreases due to the radiation losses, the displacement amplitude and mass can increase in a shoaling wave at a finite distance due to the decrease of total depth. In turn, the oscillating tail reveals a complicated behavior that includes, in particular, formation of nonlinear

wave packets as in Grimshaw and Helfrich (2008), followed by an even more complex evolution. Another important point of bifurcation is that of wave passage through the condition $h_1 = h_2$ when the polarity of a soliton is changed. According to the estimations, the interplay between the topography and rotation effects does exist under realistic oceanic conditions.

Future research should include further comparison of theory with observational results in different oceanic environments and extending the above results to the strongly nonlinear waves with rotation.

**Appendix A: Coefficients of the rotating-Gardner equation**

The eigenvalue problem for the linear long wave phase speed $c_n$ and vertical structure function for the isopycnal displacement $\Phi_n(z)$ of vertical mode $n$ is, in for the Boussinesq and rigid-lid limits (after dropping the subscript $n$),

$$\frac{d}{dz}\left[(c-\bar{u})^2\frac{d\Phi}{dz}\right] + N^2(z)\Phi = 0, \quad \Phi(-h) = \Phi(0) = 0. \tag{A.1}$$

The buoyancy frequency is

$$N^2(z) = -\frac{g}{\rho_0}\frac{d\bar{\rho}}{dz}, \tag{A.2}$$

where $\bar{\rho}(z)$ is the background density profile, $\bar{u}(z)$ is a background current in the direction of wave propagation, $g$ is the gravitational acceleration, and $\rho_0$ is a reference density. From Grimshaw et al. (2004) the coefficients of the rG equation (1) for a particular mode are given by

$$I = 2\int_{-h}^{0}(c-\bar{u})\Phi'^2dz, \tag{A.3a}$$

$$\alpha = I^{-1}\int_{-h}^{0}3(c-\bar{u})^2\Phi'^3dz, \tag{A.3b}$$

$$\beta = I^{-1}\int_{-h}^{0}(c-\bar{u})^2\Phi^2dz, \tag{A.3c}$$

$$Q = c^2I, \tag{A.3d}$$

$$\gamma = I^{-1}f^2\int_{-h}^{0}\left(\Phi'^2 - \frac{\bar{u}'}{c-\bar{u}}\Phi\Phi'\right)dz. \tag{A.3e}$$


When $\bar{u} = 0$, $\gamma = f^2/2c$, where $f$ is the Coriolis frequency. The general relation for $\gamma$ with $\bar{u}' \neq 0$ was derived by Alias et al.
(2014). The coefficient of the cubic nonlinear term is given by

$$\nu = I^{-1} \int_{-h}^{0} \left[ 3(c - \bar{u})^2 \left( 3T_z - 2\Phi'^2 \right) \Phi'^2 - \alpha^2 \Phi'^2 + \alpha(c - \bar{u}) \left[ 5\Phi'^2 - 4T_z \right] \Phi' \right] dz, \tag{A.4}$$

with the nonlinear correction to the vertical structure function, $T(z)$, found from

$$\frac{d}{dz}\left[ (c - \bar{u})^2 \frac{dT}{dz} \right] + N^2 T = -\alpha \frac{d}{dz}\left[ (c - \bar{u}) \frac{d\Phi}{dz} \right] + \frac{3}{2}\frac{d}{dz}\left[ (c - \bar{u})^2 \left( \frac{d\Phi}{dz} \right)^2 \right] \tag{A.5}$$

with $T(-h) = T(0) = 0$. The solution $T(z)$ is normalized through the addition of $b\Phi(z)$ with $b$ chosen so that $T(z_{max}) = 0$,
where $\Phi(z_{max}) = 1$. This gives the isopycnal vertical displacement $\xi(x,z,t) = \eta(x,t)\Phi(z) + \eta^2(x,t)T(z)$.

In a two-layered stratification with depths $h_1$ and $h_2$ of the upper and lower layers, respectively, and $\bar{u} = 0$,

$$c^2 = g'\frac{h_1 h_2}{h_1 + h_2}, \ \alpha = \frac{3c}{2}\frac{h_1 - h_2}{h_1 h_2}, \ \nu = -\frac{3c}{8}\frac{\left( h_1^2 + 6h_1 h_2 + h_2^2 \right)}{(h_1 h_2)^2}, \ \beta = \frac{c}{6}h_1 h_2, \ \gamma = \frac{f^2}{2c}, \ Q = 2g'c. \tag{A.6}$$

Here $g' = g\Delta\rho/\rho_0$ and $\Delta\rho = \rho_2 - \rho_1$ is the difference in densities between the lower and upper layers.

### Appendix B: Obregon et al. (2018)

For $\nu < 0$ ($0 < B < 1$), Obregon et al. (2018) (denoted ORS) equations (2.7) and (2.8) are identical to (25) and (29) using the
ORS scalings (1.4), the definitions $\mu = \nu A_0/\alpha$ (not to be confused with $\mu$ above), $\epsilon = \gamma L_0^2/\alpha A_0$, and $Ur = \alpha A_0 L_0^2/\beta$ (found
just after ORS (1.5)), and the identity

$$\tanh^{-1}\sqrt{\frac{1 - B}{1 + B}} = \frac{1}{4}\ln\left( \frac{1 + \sqrt{1 - B^2}}{1 - \sqrt{1 - B^2}} \right).$$

Note that the ORS variables for wave amplitude, $U_0$, quadratic coefficient, $\alpha$, and cubic coefficient, $\alpha_1$, have been replaced
with our notation $A_0$, $\alpha$, and $\nu$, respectively. Additionally, our spatial and their temporal decay scales are linked by $c$.

ORS (2.15) contains an error associated with in the choice of the scaling length $L_0$. There are two choices and one was used to
evaluate $\epsilon\tau_{ext}(B_0)$ and the other to define $Ur$. First consider their evaluation of the rKdV equation extinction scale which they
take as $\tau_{ext-O} = \epsilon^{-1}$ (our notation $-O$ in the subscript), or in dimensional variables, $T_{ext-O} = 1/(\gamma L_0) = \gamma^{-1}(\alpha A_0/12\beta)^{1/2}$
$(= X_{eO}/c)$. This correct result is obtained using $L_0 = \Delta_{KdV} = (12\beta/\alpha A_0)^{1/2}$, where $\Delta_{KdV}$ is the width scale of the KdV
solitary wave. This choice for $L_0$ gives $Ur = \alpha A_0 L_0^2/\beta = 12$. However, in deriving ORS (2.15) they take $Ur = 24/(1 + B_0)$,
which is correct for $L_0 = \Delta_G = \Delta_{KdV}\sqrt{2/(1 + B_0)}$, where $\Delta_G$ is the Gardner solitary wave width scale. Thus ORS apply
(2.15) using $L_0 = \Delta_{KdV}$ to evaluate $\epsilon\tau_{ext}(B_0) = \tau_{ext}(B_0)/\tau_{ext-0}$ and $Ur$ defined from $L_0 = \Delta_G$. When consistently using
either $L_0$-$Ur$ pair the corrected version of ORS (2.15) has the prefactor $\sqrt{8/(1 - B_0)}$ on the right side and is equivalent to
(28) for $0 < B_0 < 1$. Their inconsistent parameter definitions persists in the analysis for $\nu > 0$.

*Author contributions.* The authors contributed equally to this work.





*Competing interests.* There are no competing interests.

*Acknowledgements.* KRH was supported by Grant N00014-18-1-2542 from the Office of Naval Research and Grant OCE-1736698 from the National Science Foundation.



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
