# Peer review of "Effects of Rotation and Topography on Internal Solitary Waves Governed by the Rotating-Gardner Equation"

_Nonlinear Processes in Geophysics, 2022_

## Referee Comment (RC2)

**Review of "Effects of Rotation and Topography on Internal Solitary Waves Governed by the Rotating-Gardner Equation"**

This is a very well written manuscript, which develops adiabatic theory for the rotating–Gardner equation and compares this with well chosen numerical simulations, which demonstrate that the combined effects of slowly varying depth, high order nonlinearity and rotation are significant for internal waves propagating in realistic oceanic conditions. Some minor comments on the manuscript follow:

- Some of the equations on p. 4 are missing punctuation.

- Line 86, Generally would not start line with an equation number. Would suggest changing that to "Equation (9)".

- Label on Figure 1 should be $X_{eG}/X_{e0}$. Same mistake on lines 186 and 187.

- Line 211, Should omit "first" at the end of this line, as this appears to be the only model that is considered.

- Line 254–256. When discussing Figure 6(b) it is would be helpful to note that the region of integration for calculating the adiabatic response only includes the leading solitary wave and not the secondary solitary wave which develops from the trailing radiation.

---

## Author Comment (AC1)

Reply to the Reviewer 1: The authors would like to thank the reviewer for his/her careful review. Our responses to the Reviewer are indicated below by the blue font sections inserted within the original text of the review. We have also used blue font in the revised manuscript to indicate all changes to aid in this latest version.

**RC1**: 'Comment on npg-2022-3', Anonymous Referee #1, 10 Mar 2022  reply
The review of the manuscript

"Effects of Rotation and Topography on Internal Solitary Waves Governed by the Rotating-Gardner Equation"

by Karl R. Helfrich and Lev Ostrovsky

This referee was asked to join the discussion at the late stage, when the previous comments were no longer visible on the system. Hence, my comments are completely independent of the previous discussion which I have not seen.

The paper continues the line of research on internal waves within the scope of the KdV-type models. This study is based on the use of the variable-depth, rotating Gardner model, i.e. it aims to highlight the effects due to the common action of nonlinearity (including cubic nonlinearity), dispersion, rotation and depth inhomogeneity. Analysis is based on the previously developed adiabatic theory (with some corrections) and is compared to the results of direct numerical simulations of this model equation using a pseudospectral method. The results presented in a number of high-quality Figures look instructive and convincing, showing good agreement for the case of weak nonlinearity, and clarifying the limitations of the adiabatic theory when the amplitude parameter is increased.

In the view of this referee, the following points need to be clarified in order to put the research in the context of other studies:

1. Could you, please, provide a reference to the derivation of this model equation from the primitive equations. Can it be derived in a systematic way? Does this model include all terms appearing at the relevant order of the amplitude parameter?

If not, please, explain why and when we can disregard some terms appearing in the systematic asymptotic derivation (e.g., the 5th derivative and nonlinear-dispersive terms in the extended KdV equation). I suspect that there could be some additional terms in the case of variable topography and rotation.

You probably mean equation 1 in our manuscript. This equation without the cubic term is derived in detail in the paper by Grimshaw, Guo, Helfrich, and Vlasenko (JPO, 44, 1116, 2014) which is cited in our manuscript. A similar equation with both quadratic and cubic nonlinearities without inhomogeneity is discussed in many papers, including Obregon, Raj, and Stepanyants (Chaos, 28, 033106, 2018) which is also cited in our manuscript; it also references some previous works in this area.   Note that classical derivation of unidirectional evolution equations, beginning from KdV, is based on the smallness of all terms added to the

basic operator $\partial/\partial t + c\partial/\partial x$ so that at a given order, other terms are additive and in general are considered of the same order. If the latter are, in turn, of different orders, more terms (such as the one with the 5[th] derivative) can play a significant role. Note that non-additive, nonlinear-dispersive terms can be important for strongly nonlinear variants of similar evolution equations (e.g., Ostrovsky and Grue, Phys. Fluids,15, 2934, 2003). Lastly, adding just the cubic term to the quadratic one in the KdV model makes a significant change only when these two nonlinear terms are of the same order. This can be achieved, for example, when the layer depths (in the two-layer version) are nearly equal. However, it has been demonstrated in numerous studies that the Gardner equation has good oceanic applicability even in situations when the wave amplitude is not small. Thus, it has become the phenomenological model of choice. This had already been stated in the second paragraph of the original submission of the manuscript. Now we have slightly extended the discussion and included three references on this aspect (Stanton and Ostrovsky, 1998; Pelinovsky et al., 2000; Grimshaw et al., 2004). See the new 3[rd] paragraph of the Introduction.

See also the answer to the next question.

Also, please, compare to the results in this paper and discuss in that context:

Karczewska, P. Rozmej, Can simple KdV-type equations be derived for shallow water problem with bottom bathymetry? Comm. Nonlin. Sci. & Numer. Sim. 82 (2020) 105073.

The work by Karczevska and Rozmej considers the analogs of KdV and Boussinesq equations for shallow-water waves with different relative orders of the small perturbing factors: nonlinearity, dispersion, and bottom slope. In such cases, naturally, different orders of these parameters can be comparable. Actually, this is the case of Gardner equation: as mentioned, the cubic nonlinearity is important when the quadratic one is small (in the oceanic case, when the layers have a close thickness); otherwise, the quadratic effect would prevail. Another example is the case when the main, third-order dispersion term is small, and we have the known Kawahara equation. That is, KdV with the 5[th] order dispersion term added. Here we consider a certain (although rather general) physical problem when all included terms are of the same order, and the classical description is sufficient. Upon revision, we added a reference to the Karczevska and Rozmej paper with a short comment in the Introduction section.

2. The results of the study need to be compared and contrasted to the two recent papers by Yury Stepanyants:

- Yu. A. Stepanyants, The effects of the interplay between the rotation and shoaling for a solitary wave on variable topography, Stud. Appl. Math. 142 (2019) 465-486.

That paper was published in parallel with our paper by Ostrovsky and Helfrich (2019) and the authors have been in contact with Yu. Stepanyants (see the mutual acknowledgments in these papers). In the two papers, we concentrated on different aspects of the model, its limitations, and its applications to oceanic conditions. Indeed, in his paper Stepanyants mentioned a case when rotation and inhomogeneity effects can be balanced for the soliton amplitude. It is a very

special case of parameter choices; moreover, there is still no such balance for fluid velocity and, more important, soliton energy, which still decreases due to radiation. We have added a short paragraph in Section 4.2 (lines 227-230) on this point.

- Y.A. Stepanyants, Nonlinear waves in a rotating ocean (the Ostrovsky equation and its generalizations and applications), Izvestiya, Atmospheric and Oceanic Physics, 56 (2020) 20-42.

This paper is a review of the problem based on previous publications. If the reviewer wants us to mention a specific effect, please let us know.

In particular, in these studies, it was shown that for solitary waves moving towards shallower waters, the terminal decay caused by rotation can be suppressed by the shoaling effect. Is this result confirmed in your studies?

See the answer above. Again, the soliton energy always decreases due to rotation.

3. This sentence in the Introduction is only partially correct:

"It is also known that rotation destroys internal solitons due to resonant radiation of inertia-gravity waves (terminal damping; see Grimshaw at al., 1998a)."

This is true in the absence of currents. However, the sign of the rotation coefficient in the Ostrovsky equation may be changed by the underlying shear flow, and then the equation supports solitons, see the first examples in

A. Alias, R.H.J. Grimshaw, and K.R. Khusnutdinova, Coupled Ostrovsky equations for internal waves in a shear flow, Phys. Fluids 26 (2014) 126603.

This is true. There exist examples of stratification producing negative dispersion and, correspondingly, non-damping solitons with rotation. We added a short discussion to the Introduction section in lines 21-23.

4. Lines 19-20, should it be 'specific case'?

Yes, the text has been corrected to include "specific case."

To summarise, assuming that the model equation can be justified, the study makes perfect sense, and the results are convincing. The validity of the model equation is not entirely clear and needs to be clarified by putting the research in the context of some other studies in the field, as suggested above.

Lastly we note for the reviewer that we have removed the second Appendix that had discussed the results by Obregon et al (2018).

---

## Author Comment (AC3)

Reply to reviewer 2: The authors would like to thank the reviewer for the quick and supportive response to our submission. We have made all of the requested changes and corrections. The revised manuscript uses blue font to indicate all changes that we have made in response to both reviewers.

Review of "Effects of Rotation and Topography
on Internal Solitary Waves Governed by the Rotating-Gardner Equation"

This is a very well written manuscript, which develops adiabatic theory for the rotating–Gardner equation and compares this with well chosen numerical simulations, which demonstrate that the combined effects of slowly varying depth, high order nonlinearity and rotation are significant for internal waves propagating in realistic oceanic conditions. Some minor comments on the manuscript follow:

- Some of the equations on p. 4 are missing punctuation.
- Line 86, Generally would not start line with an equation number. Would suggest changing that to "Equation (9)".
- Label on Figure 1 should be $X_{eG}/X_{e0}$. Same mistake on lines 186 and 187.
- Line 211, Should omit "first" at the end of this line, as this appears to be the only model that is considered.
- Line 254–256. When discussing Figure 6(b) it is would be helpful to note that the region of integration for calculating the adiabatic response only includes the leading solitary wave and not the secondary solitary wave which develops from the trailing radiation.